# Texture-Less Shiny Objects Grasping in a Single RGB Image Using Synthetic Training Data

Chen Chen [1] , Xin Jiang [1,*] , Shu Miao [1], Weiguo Zhou [2] and Yunhui Liu [1,2,*]

1   Mechanical Engineering and Automation, Harbin Institute of Technology, Shenzhen 518055, China; chenchen92@stu.hit.edu.cn (C.C.); Shu.Miao@hotmail.com (S.M.)
2   Department of Mechanical Engineering, The Chinese University of Hong Kong, Hong Kong, China; weiguozhou@cuhk.edu.hk
*   Correspondence: x.jiang@hit.edu.cn (X.J.); yhliu@mae.cuhk.edu.hk (Y.L.)

**Abstract:** In the industrial domain, estimating the pose of texture-less shiny parts is challenging but worthwhile. In this study, it is impractical to utilize texture information to obtain the pose because the features are likely to be affected by the surrounding objects. In addition, the colors of the metal parts are similar, making object segmentation challenging. This study proposes dividing the entire process into three steps: object detection, feature extraction, and pose estimation. We use the Mask-RCNN to detect objects and HRNet to extract the corresponding features. For metal parts of different shapes, different keypoints were chosen accordingly. Conventional contour-based methods are inapplicable to parts containing planar surfaces because the objects occlude each other in clustered environments. In this case, we used dense discrete points along the edges as semantic keypoints for metal parts containing planar elements. We chose skeleton points as semantic keypoints for parts containing cylindrical components. Subsequently, we combined the localization of semantic keypoints and the corresponding CAD model information to estimate the 6D pose of an individual object in sight. The implementation of deep learning approaches requires massive training datasets and intensive labeling. Thus, we propose a method to generate training datasets and automatically label them. Experiments show that the algorithm based on synthetic data performs well in a natural environment, despite not utilizing real scenario images for training.

**Keywords:** synthetic training data; shiny object pose estimation; single RGB image

## 1. Introduction

Industrial metal parts are necessary components of many products. Therefore, it is essential to facilitate computers to recognize and locate industrial metal parts in tasks like pick-and-place and assembly. Estimating their pose is vital for robotic grasping. However, this issue is challenging because of the dynamic changes in the representation of metal parts. The surface of metal parts is texture-less and shiny, which poses a considerable challenge for multiple traditional pose estimation approaches, owing to their dependence on reliable visual features. The illumination, occlusions, and clutter between objects may affect the actual appearance of the industrial parts. Figure 1 shows the challenges of estimating the pose of shiny metal parts: the light may cause the metal parts to have specular highlights; the cluttered metal parts may occlude with each other; metal parts with low-light may drive extreme darkness and the metal parts may induce interreflection with others.

Some pose estimation approaches (e.g., [1,2]) require the depth information retrieved by actively projecting coded patterns, which are ineffective for shiny metal parts because the projected patterns may have a misleading reflection or may be undetectable. Thus, designing a strategy that does solely depend on RGB images is essential.

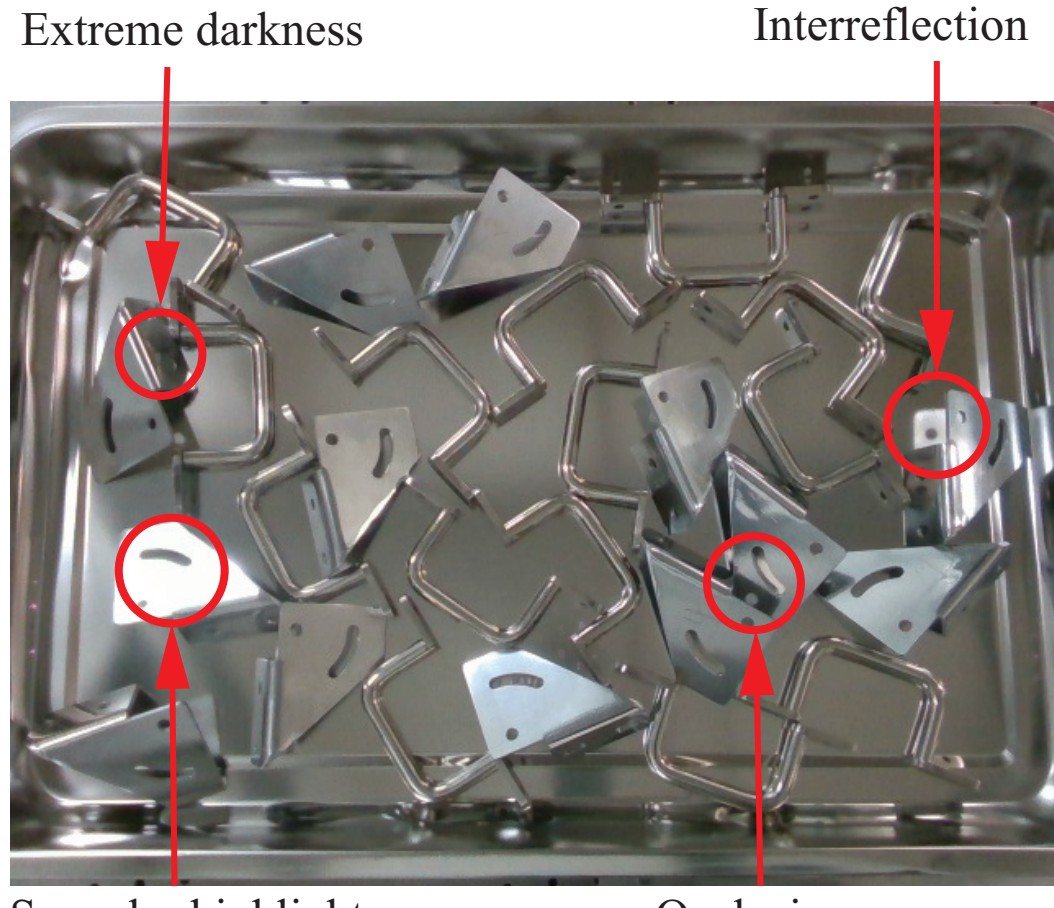

**Figure 1.** Challenges of Pose Estimation for Shiny Metal Parts.

Traditionally, researchers estimate the 6D object pose using object features (e.g., keypoints, color and silhouette). In robotic bin-picking applications, the perception of textureless, highly reflective parts is a valuable but challenging task. The high glossiness can introduce fake edges in RGB images and inaccurate depth measurements [3]. The active lighting strategies may improve the pose estimation performance. Still, the metal parts are stacked together, and the lights have to change according to the surroundings of the metal parts, so it is not robust to the problem. The active lighting strategies will lead to extra cost. In addition, in some application scenes, like sorting surgical tools, the background is a shiny metal plate which poses a challenge to the estimation problem. Last, the labeled data is not only used in the object detection but also keypoint detection. It is difficult to label keypoint accurately in the real data, so we propose a method to use synthetic images with labeled keypoints. Hence, it retains the limitations of texture-less metal parts. Recently, deep learning has proved its effectiveness among computer vision techniques because it is trained on real images or synthetic data. It is therefore more accurate and robust than traditional feature-based methods. However, it is influenced by training data which influence their proformances and require intensive manual labor to mark the datasets. The OHEM [4] is a good algorithm for object detection. Still, it is not suitable for the keypoints detection since we need to label the semantic keypoints accurately, which is a challenging task for real images. To overcome this problem, some researchers have used synthesizing datasets [5–7] and produced ground truth labels [8,9]. Some researchers use unlabeled data [10,11] in object detection, but the methods are not applicable to our keypoints detection problem.

Using deep learning approaches to estimate the 6D pose of an object requires training samples, which include objects with accurate annotations. It is expensive to annotate

them using human labor. To solve this problem, we propose a novel strategy for 6D pose estimation using synthetic training data. The proposed system takes an RGB image as input to detect shiny metal parts, and subsequently uses semantic keypoint detection to extract the features, a novel method for interpreting the detected edges or skeleton in the semantic keypoints detection. Particularly, our approach does not count on continuous contours, but explores distinct combinations of dense discrete points along the edges. We developed an efficient method to automatically synthesize many shiny metal part datasets using Blender. By changing the background and the illumination, our dataset can cover diverse conditions and ensure the robustness of the trained neural network. We designed an algorithm to automatically label the training data. In this way, we can bypass the effect of the uncontrollable aspects (e.g., human fatigue and distraction) and guarantee the accuracy of the labeling compared with manual annotation. According to a previous study [9], sounder labeling precision may result in a better performance.

We have several contributions to tackle the study of 6D pose estimation of clustered shiny metal parts. We proposed a system to compute the shiny object pose. In this framework, we use the object detection method to detect the shiny metal parts, then use semantic keypoints detection to extract the features. We developed an efficient way to synthesize many shiny metal parts datasets using Blender automatically. We designed an algorithm for automatically labeling the training data, which can bypass the effect of uncontrollable aspects and then guarantee the accuracy of the labeling compared with manual do it.

In this study, we sought to estimate the pose of shiny industrial parts. Our approach employs synthesized images as training data. The key ideas and corresponding work in this area are discussed below.

Estimating the pose of shiny industrial parts using a single image is a prevalent and essential topic. Although pose estimation has achieved excellent performance on textured objects, its performance on texture-less shiny metal parts remains poor. The difficulty in extracting exact depth information from reflective surfaces is a challenge using RGBD-based approaches. However, RGB-based methods also have limitations. The mechanism of correspondence-based methods generally involves a known object model and 2D keypoints corresponding to the 3D coordinates [12]. This type of issue is usually called Perspective-n-Point issue. These methods typically employ feature descriptors (e.g., SIFT [13]). However, these methods fail to address the poses of the texture-less objects. Some researchers have used template-based methods for texture-less objects [14]. These methods collect a template set of poses and corresponding images. Subsequently, they compared it with the target image using the extracted features, such as gradient information. However, these techniques cannot achieve high accuracy because the samples are discrete. Thus, most strategies, such as those of Hinterstoisser et al. [14] and Hodan et al. [15], require further improvement using depth information. Only a few techniques based on RGB images, such as those proposed by He et al. [16], can be used for metal parts. However, they have limitations in terms of the geometric shapes of the metal parts. Recent publications regarding pose estimation have employed deep learning techniques that have achieved remarkable improvements compared with traditional methods. Voting-based approaches, such as PVNet [17], attempt to classify every pixel using a voting vector before calculating the pose. Some regression-based approaches, such as PoseCNN [18], directly output a pose representation, whereas others, such as SSD-6d [19] and BB8 [20], employ an architecture for estimating 2D coordinates and then recovering the 6D pose of the object. Zhang et al. [21] employed edge details to estimate a 6D pose. Although deep learning methods work well on public benchmarks (e.g., T-LESS [2]) for ordinary objects, they have restrictions for the measurements of the metal parts. Moreover, it is challenging to supply each object with real image training data.

Recent leading object detection methods rely on neural networks. Nevertheless, they require large amounts of training data to perform satisfactorily. Labeling data is a costly and time-consuming task. Therefore, using synthetic images is a very attractive

concept. However, typical synthetic image rendering approaches cannot fully convey actual characteristics of real-world environments. There were gaps between synthetic and authentic images, as described in [22]. NVIDIA proposed a method [23] to bridge this gap. The synthesized data can perform well in the pose estimation area using this data generation. However, this technique cannot be used to model highly reflective metallic industrial parts.

Many cutting-edge pose-estimation approaches employ supervised learning methods. However, annotations in datasets require costly labor, and the precision of the ground truth in datasets cannot be guaranteed. LabelMe is an open-source annotation method for pixel-wise labeling. However, the annotation procedure is manual. EasyLabel [9] is a semiautomatic annotation method for labeling accurate object masks. To the best of our knowledge, there are no RGB image datasets that can automatically annotate masks. Annotating the ground truth of a target object pose is a significant task. For most target object recognition studies, such as, target classification or target detection, researchers typically annotate the ground truth manually. To assist in research pose estimation, a vast dataset with precisely annotated data is required. PASCAL3D+ [24] is a prominent dataset containing keypoint annotation. They created a MATLAB annotation tool for labeling the positions of the keypoints. In this annotation procedure, by loading the model of the target and corresponding annotated 3D points in this model, we labeled the related 2D keypoints in the images. All annotation work is performed manually, and accuracy cannot be ensured.

## 2. Materials and Methods

Figure 2 illustrates the outline of the proposed workflow. The workflow mainly contains three steps. First, we use a target object detection network to detect the metal parts. Second, we use a keypoints detection network to estimate the keypoints in the detected metal parts region. We use the same synthesized datasets to train the two networks. The datasets contain synthesized images, labeled masks, and keypoints. In the end, with the estimated keypoints information, the 6D pose of each metal part can be achieved. We employ the off-the-shelf network Mask-RCNN [25] to detect the metal parts. We use Blender to synthesize training data automatically for the network. We propose a new method to label the object's mask automatically because labeling those masks in the network requires intensive labor. Inspired by the study of using semantic keypoints in human pose estimation, we use the high-resolution representative network [26] to predict the semantic keypoints of industrial parts. In order to avoid overwhelming labor manual labeling, we label the keypoints automatically. We build the RGB datasets with synthesized images containing labeled keypoints on the corresponding object. We use the predicted object's keypoints to estimate the pose of the corresponding metal parts. The mistakenly estimated position of detected keypoints may incur the pose estimation error. In order to tackle this problem, we use Perspective-n-Point combined with RANSAC to clear the outliers.

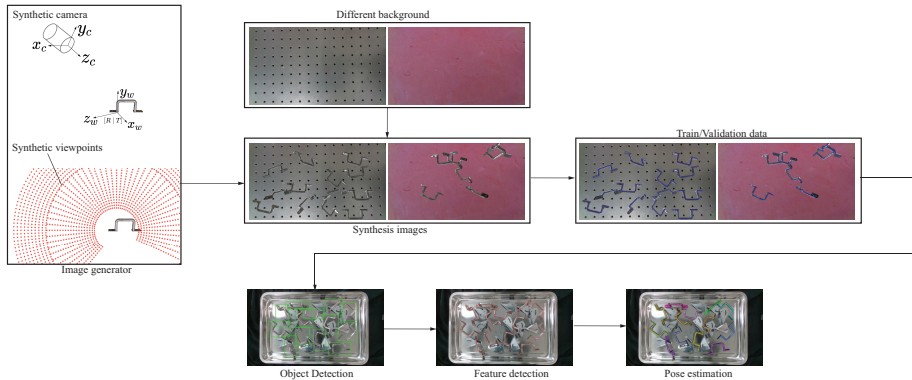

**Figure 2.** Flowchart for constructing a pose estimation framework.

### 2.1. Object Detection

We employ the Mask R-CNN network to implement object detection because it is challenging to segment metal parts traditionally. We utilize the ResNet50 as the backbone for extracting features. We utilize the synthesis data to train this metal parts detection network. Using synthetic data can detour overfitting to a specific distribution presented in the dataset. Synthetic data should wrap diverse conditions, including background deviation, light modification, and reflectance effects. The render engine Blender can simulate these conditions. This software renders the synthetic data by setting the CAD model in a virtual 3D environment with physical conditions. Figure 2 shows the setup of the image generator.

We employ various real world scenes as the background of a virtual environment and place the CAD model in diverse positions and orientations to generate a series of images portraying the metal parts with various backgrounds and poses. The Figure 3 shows the rendered synthetic images with different backgrounds and illumination.

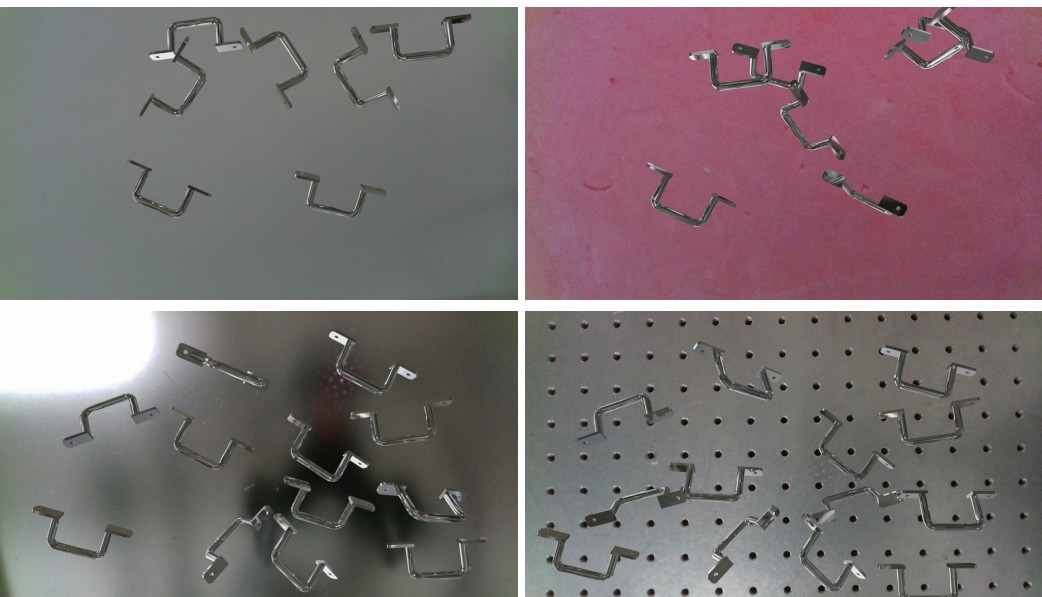

**Figure 3.** The generated synthetic images with different backgrounds and illumination.

After creating the synthesized dataset, we find a method for automatically labeling the pixel-wise mask. In this labeling procedure, the principal idea of mask labeling is to use Blender for rendering images without background information. Then we set the model with non-metallic material property in Blender, for example, plastics. The second image in Figure 4 illustrates the result.

We binarize each metal part separately. Then we discover the silhouette of each part and fill it. The filled area is the mask. The demanding aspect is about finding the silhouette of per metal part. Since the color of per part appears distinct with diverse lighting situations and positions, it is not easy to detect their silhouettes directly. We employ an intermediate image to label, as illustrated in Figure 4. If we set the model's property as non-metallic, like plastic, its image appearance may be robust to lighting conditions. After using different colors to set CAD models, we can discover the silhouette for each instance. Figure 4 illustrates this.

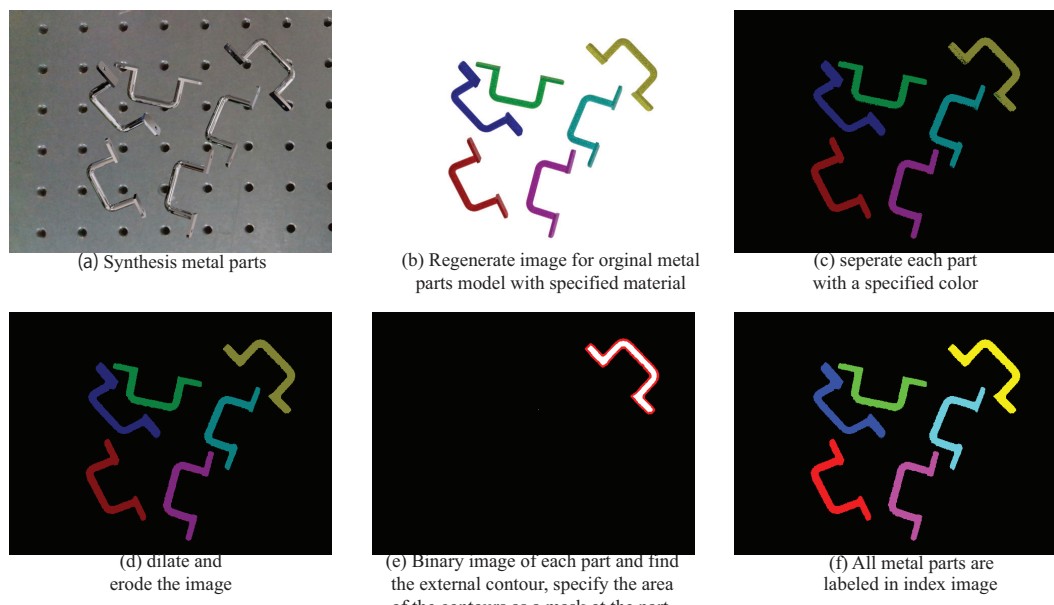

(a) Synthesis metal parts

(b) Regenerate image for orginal metal parts model with specified material

(c) seperate each part with a specified color

(d) dilate and erode the image

(e) Binary image of each part and find the external contour, specify the area of the contours as a mask ot the part.

(f) All metal parts are labeled in index image

**Figure 4.** The procedure of annotating pixel-wise labels, mainly containing six stages.

## 2.2. Semantic Keypoints Detection

Inspired by the High-Resolution Network (HRNet) architecture used in human pose estimation [26], We utilize the HRNet as a feature detector for detecting the semantic keypoints of the metal parts. The HRNet sustains high-resolution representations in the entire method. The network can connect the high-to-low resolution convolution streams in parallel and repeatedly exchange the infomation across resolutions. By this method, the resulting representation is semantically richer and spatially more precise. This module contains high to low along with low to high resolution. Then the global and local feature information can be received and merged to decide the position of the keypoints. The resulting network consists of 4 stages.

We use the confidence maps ('heatmaps') as the way to describe the keypoints. The Figure 5 illustrates the heatmap responses of the HRNet module. Given an input image, a spatial ConvNet regresses joint heatmaps for the image. The network is trained to regress the location of the metal parts joint positions. However, we regress a heatmap of the keypoint positions separately in an input image instead of straight regressing the keypoints $(x, y)$ positions. The output of the last convolutional layer is a fixed-size; here it is $64 \times 64 \times k$ for $k$ is number of joints.

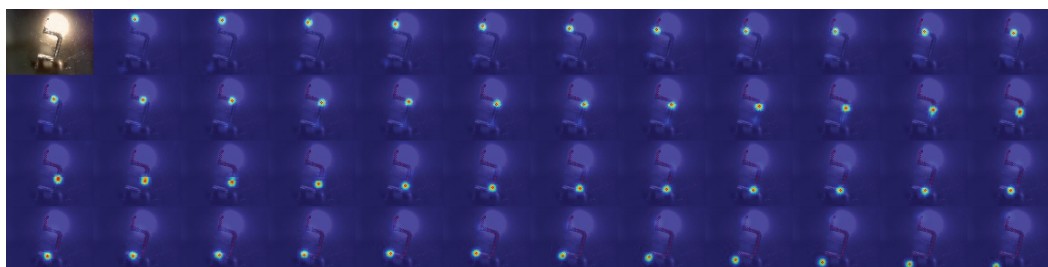

**Figure 5.** This figure illustrates the heatmap responses of the HRNet module. The upper left is the raw image, and the rest are heatmaps of each keypoints.

At training time, the ground truth labels are heatmaps synthesised for each joint separately by placing a Gaussian with fixed variance at the ground truth joint position. We use the $l_2$ loss to penalise the squared pixel-wise differences between the predicted heatmap and the synthesised ground truth heatmap.

We can use the CAD model to create the image of the corresponding object alone without background when generating the synthesis image in the PNG format. We divide the PNG synthesis image information into Red, Green, Blue, and alpha channels. We can use the alpha channel in the synthesis image to judge whether the pixel belongs to the metal part. Then we can get the bounding box of the object.

It is challenging for humans to label semantic keypoints accurately. Thus, we suggest automatically labeling them in synthesis images. We should choose keypoints that reflect the metal characteristics. We choose the discrete points along the edge as the semantic keypoints for the plate surface. For the curved or symmetrical metal parts, we choose the discrete points along the skeleton of the parts.

The dataset includes diverse kinds of photos in different scenarios. We use the photographs taken from the natural world as the background. Then we can develop many synthetic images with various backgrounds. In this way, we can acquire different types of training data with annotated semantic keypoints. In Blender, by setting the intrinsic camera parameters of the virtual camera, we can acquire different cameras. In creating metal parts datasets, this process comprises the following steps: loading CAD models, determining keypoints, and labeling them. Figure 6 exhibits the results of different metal parts keypoints labeling.

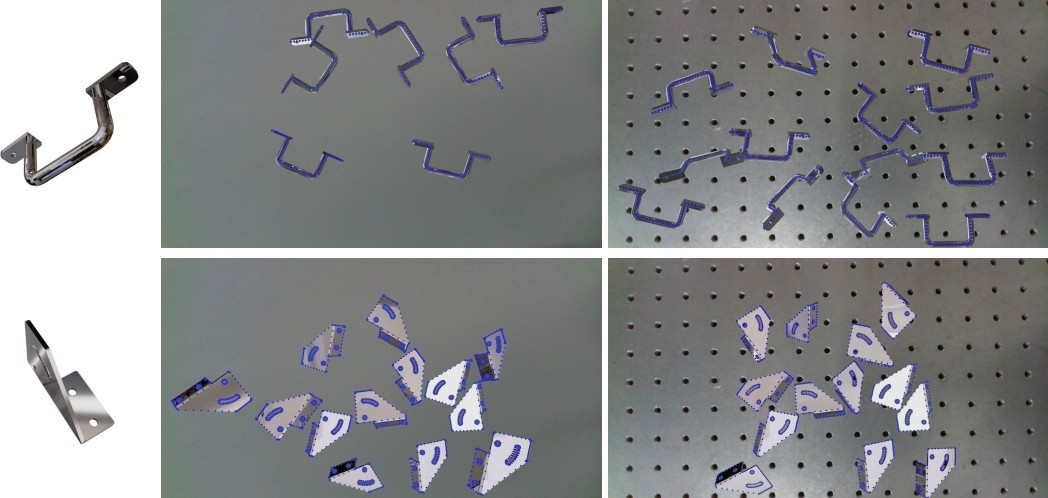

**Figure 6.** The figure illustrates the results of labeling keypoints. The first column is the metal model, the second and third columns are the synthesis images with keypoints. The blue dots are the marked keypoints.

*2.3. Pose Estimation*

As for the industrial parts, we can get their CAD models quickly from mechanical design software. Once we obtain the model, we can get the 3D positions of the specified keypoints from the corresponding model. Then we can get their corresponding 2D projection positions of the keypoints in the images. Using the HRNet network module can get the predicted positions of those specified corresponding keypoints. However, some predicted keypoints are imprecise due to some incorrect detections. We offer to utilize the PnP and RANSAC to extract the outliers and then get the pose to handle this issue.

**3. Results and Discussions**

In this section, we develop a program to implement the proposed method and conduct experiments to verify its feasibility. The Figure 7 shows the experimental setup. This section estimates the pose of two kinds of metal parts. We use a Realsense D435 camera to obtain the RGB image, and then we employ a network to detect the objects. After the object is detected, we utilize HRNet to get semantic keypoints. Then we use the keypoints and the

corresponding 3D points information to estimate the pose. Finally, we use the UR5e robot to grasp the metal parts.

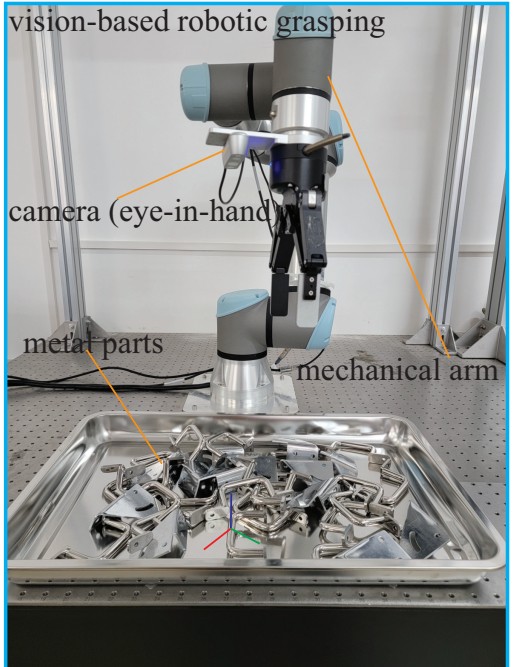 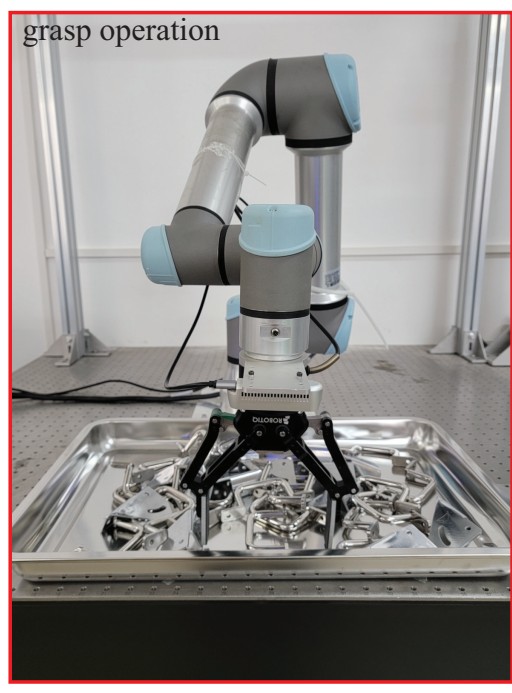

**Figure 7.** Experiment setting: the setup is a grasping metal setting.We use the eye-in-hand camera to capture the RGB image and drive the robot arm to grasp the metal parts.

### 3.1. Object Detection Performance

We use the Blender rendering engine to create 20,000 images. Each photo includes several industrial parts. All the photos have different backgrounds. The Mask-RCNN [25] is used as the detection network. We employ synthetic images to train the network. We train a ResNet-50-FPN model that shares features between the RPN and Mask R-CNN stages. Mask R-CNN is fast to train. Training with ResNet-50-FPN on METAL datasets takes 32 h in our synchronized 4-GPU implementation (0.72 s per 16-image mini-batch). The backbone is the resnet with fpn. The depth is 50. Each batch contains 16 images. We use the Pytorch deep learning framework to train the model, the size of the parameter is 351.1 MB. The green bounding box in the second column of Figure 8 depicts the metal parts detection results.

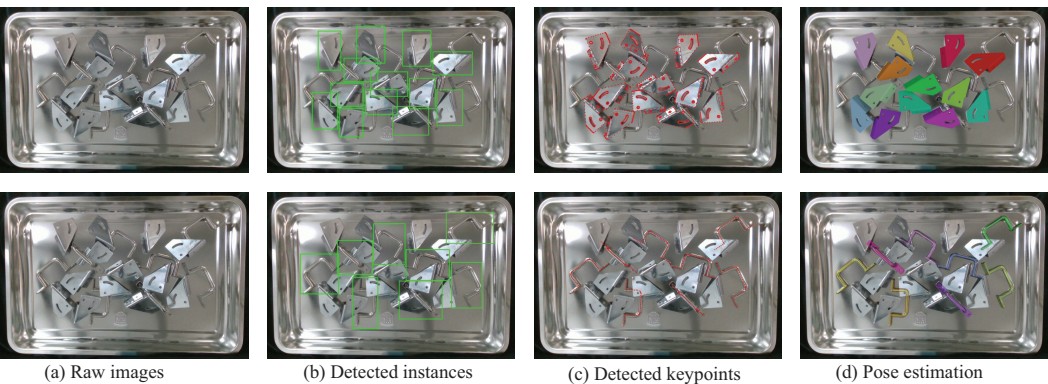

(a) Raw images    (b) Detected instances    (c) Detected keypoints    (d) Pose estimation

**Figure 8.** Pose estimation process: the first column images are the raw images, the second column images are the detected shiny industrial parts marked with the green bounding box, the third column images are corresponding predicted keypoints marked with red dots, the last column images are the 3D model with estimated poses overlaid in a specified color.

### 3.2. Semantic Keypoints Detection Performance

We use the top-down paradigm similar to [26]. It detects the metal object instance using an object detector and then predicts keypoints. We train HRNet-W48 from scratch with input size $384 \times 288$. We extend the height or width of the metal parts bounding box into a fixed aspect ratio: height:width = 4:3. We clip the bounding box from the image and resize it to the specified size $384 \times 288$. We choose keypoints on the metal CAD model and project the 3D keypoints to the 2D images for training and testing in the deep high-resolution neural network. We employ 20,000 images containing metal parts with various rotational and translational positions. Figure 8 shows the result. The confidence of the detection box determines the order of predicting objects. Better object detection confidence implies the better recognize corresponding object.

We use the Pytorch as a deep learning framework in the keypoints detection part. We use four Tian Xp for training. The base learning rate is $5 \times 10^{-4}$. For the circular metal parts, we choose 47 keypoints as the semantic keypoints. In this part, we train 32 epochs. We choose 162 keypoints along the metal edges for the planar metal parts as the semantic keypoints. In this part, we train 35 epochs. Since the keypoints of metal parts cannot be accurately marked in real images, we both train and test in synthetic images. The Table 1 illustrates the results.

**Table 1.** The results of keypoints detection.

| Object Type | Method | Backbone | Input Size | $AP$ | $AP_{50}$ | $AP_{75}$ | $AP_L$ | $AR$ |
|---|---|---|---|---|---|---|---|---|
| circular | HRNet | HRNet | $384 \times 288$ | 0.640 | 0.699 | 0.642 | 0.670 | 0.694 |
| planar | HRNet | HRNet | $384 \times 288$ | 0.667 | 0.776 | 0.677 | 0.772 | 0.699 |

### 3.3. Grasping Metal Parts

The last column of Figure 8 illustrates that we use colorful models to render the metals parts with estimated pose in the images. To test the pose estimation performance of the proposed method, we designed a part picking test. In the beginning, We piled metal parts randomly in the visual field and attempted estimating poses by the proposed approach. Once the presented algorithm obtained a metal part pose, we grasped this part and repeated the procedure. The grasping of metal parts has two kinds of subjects: plate metals and circular metal parts. The Figure 9 shows the procedure of grasping two kinds of metal parts.

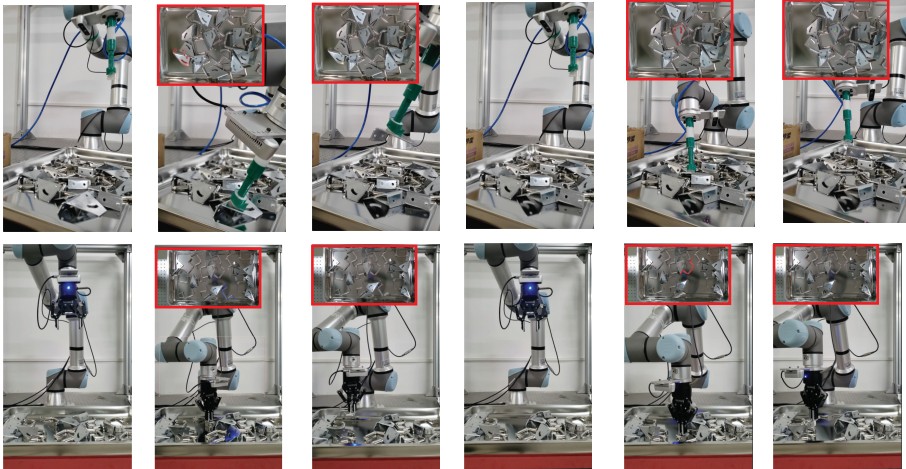

**Figure 9.** Robot grasping demonstration. Our method can segment each object instance automatically and estimate its 6D pose, then the robot grasps or sucks the recognized parts.

*3.4. Processing Time*

We tested the algorithm on a platform equipped with an Intel Core i9 7980XE and four NVIDIA GeForce GTX Titan Xp. For object detection part, it takes 0.3 s. It takes 0.1 s for keypoints detection and less than 0.01 s for pose estimation. The whole running time depends on the number of metal parts.

## 4. Conclusions

We present a new strategy for estimating the pose of clustered metal parts. We devised a new way to automatically synthesize and label datasets, then we employed the datasets to train the networks. In our experiments, the networks performed well in detecting the metal parts and predicting the corresponding keypoints. Subsequently, we successfully estimated the pose of the metal parts and validated our approach's effectiveness through robot grasping experiments. Our solution expands the genre of pose estimation for texture-less and shiny objects, exhibiting excellent value for the evolution of industrial automation.

**Author Contributions:** Conceptualization, C.C. and X.J.; methodology, C.C.; software, C.C. and W.Z.; validation, C.C. and X.J.; formal analysis, C.C.; investigation, C.C.; resources, C.C.; data curation, C.C.; writing—original draft preparation, C.C., X.J. and S.M.; writing—review and editing, C.C., X.J., S.M. and Y.L.; visualization, C.C. and X.J.; supervision, X.J. and Y.L.; project administration, X.J.; funding acquisition, X.J. and Y.L. All authors have read and agreed to the published version of the manuscript.

**Funding:** This work was supported in part by National Key Research and Development Program of China under Grant 2018YFB1309300, in part by National Natural Science Foundation of China (Grant No. 61873072), in part by Shenzhen Research Grant for Science and Technology Development (Grant No. JSGG20210420091804012).

**Institutional Review Board Statement:** Not applicable.

**Informed Consent Statement:** Not applicable.

**Data Availability Statement:** Not applicable.

**Conflicts of Interest:** The authors declare no conflict of interest.

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
