# Peer review of "Texture-Less Shiny Objects Grasping in a Single RGB Image Using Synthetic Training Data"

_applsci, doi:10.3390/app12126188_

Round 1

Reviewer 1 Report

I found the study quite interesting and timely- The manuscript is written well and organized clearly. I recommend the publication of the work. 

Author Response

The authors thank the reviewer for the positive comments on our work.

Reviewer 2 Report

In this paper, the author proposes to leverage synthesis images to train one robust network for object detection and pose estimation.  

The paper is good, and I only two main concerns as follow:

  1. The synthesised images have been widely used in computer vision. It would be good the author could discuss them.

[1] Learning from Simulated and Unsupervised Images through Adversarial Training CVPR 2017

[2] Unlabeled Samples Generated by GAN Improve the Person Re-identification Baseline in vitro. ICCV 2017

2. I am interesting whether the hard negative mining could yield similar result without synthesised images.

[3] Shrivastava, A., Gupta, A., & Girshick, R. (2016). Training region-based object detectors with online hard example mining. In Proceedings of the IEEE conference on computer vision and pattern recognition (pp. 761-769).

Reviewer 3 Report

Article "Texture-less Shiny Objects Grasping in a Single RGB Image
Using Synthetic Training Data" is interesting and current. Photographic documentation and tables are clear and well-described
I did not find anything to complain about the author of the article.

Author Response

(The authors gave the same response as above.)

Reviewer 4 Report

The authors seem to intend to propose a machine vision algorithm to identify (they call it “to grasp”) shiny (texture-less) objects and determine their position and orientation. The manuscript is not publishable.

It is not clear how this research advances the knowledge base. The machine vision task of identifying objects and estimating their pose, scale, percentage of occlusion when they are partially covered by other objects based on learned templates is a common task in any vision system such as Cognex, Fanuc iR Vision 2D, National Instruments, etc. All these systems use well established, successfully demonstrated functions for geometric matching, pattern matching and edge finding techniques used for this task. The template used to learn the model can be the image of a real object or a digital model, such as a CAD drawing of the object which is searched.

The authors state that “The surface of metal parts is texture-less and shiny, which poses a considerable challenge for multiple traditional pose estimation approaches”.  The authors seem to try to solve this aspect. However, this is not an issue in machine vision. Using correct lighting strategies, such as Bright Field or Dark Field lighting strategies make identification of shiny metal objects and their features an easy task. Also, using a correct background surface improves the task of object identification.

The authors seem to deliberately confuse two different machine vision tasks: (1) geometric or pattern matching used to identify objects, determine their pose, scale and occlusion (this is what the authors declare to be the task studied in the manuscript), and (2) object classification used to determine the appartenance of objects to various classes based on their texture, color spectrum, shapes, etc. For example, the authors mention that “Labeling data is a costly and time-consuming task” (line 95). Note that labeling data is a step used in object classification, not in geometric or pattern matching.

Round 2

Reviewer 4 Report

The manuscript is still unpublishable. The authors do not clarify how their research advances the knowledge base. The machine vision task of identifying objects based on their features using geometry or pattern matching and edge finding techniques are typically achieved by students in classroom assignments during a Machine Vision course, with the observation that unlike these authors, students know what they are doing.

For example, to my dismay, even in their response to my comments, the authors mention that “The high glossiness can introduce fake edges in RGB images and inaccurate depth measurements”. Indeed, why do these authors insist on performing object matching vision tasks using analysis of RGB images? These tasks are typically done on grayscale images after converting the RGB image into an HSL (Hue-Saturation-Luminance) image and extracting the Luminance (L) plane. The grayscale image (the Luminance plane) is then analyzed for geometry or pattern matching.

Unlike these authors, students in a Machine Vision course are aware of what imaging system they are using. These authors mention in the manuscript names of what seem to be machine vision algorithms (Mask-RCNN, HRNet, OHEM, etc.), but they do not mention, even by the way, what machine vision software they use. Is it LabView with some of its modules? Is it Matlab with one of its modules? Is it Cognex? Is it Fanuc iR Vision? Is it some open-source software?

In their answer to my review the authors declare that their scope of work has two machine vision tasks, one of them being “keypoint” detection. In machine vision parlance, what the authors call “keypoint” is called “fiducial”. Identification of fiducials for object matching and orientation is just an intermediate task typically achieved through techniques of edge finding.

I recommend that the authors send a manuscript for publication in a scientific journal after they get familiar with machine vision techniques.